# Turning microplastics into nanoplastics through digestive fragmentation by Antarctic krill

Amanda L. Dawson[1], So Kawaguchi[2], Catherine K. King[2], Kathy A. Townsend[3], Robert King[2], Wilhelmina M. Huston[4] & Susan M. Bengtson Nash[1]

Microplastics (plastics <5 mm diameter) are at the forefront of current environmental pollution research, however, little is known about the degradation of microplastics through ingestion. Here, by exposing Antarctic krill (*Euphausia superba*) to microplastics under acute static renewal conditions, we present evidence of physical size alteration of microplastics ingested by a planktonic crustacean. Ingested microplastics (31.5 μm) are fragmented into pieces less than 1 μm in diameter. Previous feeding studies have shown spherical microplastics either; pass unaffected through an organism and are excreted, or are sufficiently small for translocation to occur. We identify a new pathway; microplastics are fragmented into sizes small enough to cross physical barriers, or are egested as a mixture of triturated particles. These findings suggest that current laboratory-based feeding studies may be oversimplifying interactions between zooplankton and microplastics but also introduces a new role of Antarctic krill, and potentially other species, in the biogeochemical cycling and fate of plastic.

[1] Environmental Futures Research Institute, Southern Ocean Persistent Organic Pollutants Program, Griffith School of Environment, Griffith University, 170 Kessels Road, Nathan, QLD 4111, Australia. [2] Australian Antarctic Division, Department of the Environment and Energy, 203 Channel Highway, Kingston, TAS 7050, Australia. [3] School of Biomedical Sciences, Moreton Bay Research Station, University of Queensland, North Stradbroke Island, QLD 4183, Australia. [4] School of Life Sciences, University of Technology Sydney, Faculty of Science, 15 Broadway Ultimo, Sydney, NSW 2007, Australia. Correspondence and requests for materials should be addressed to A.L.D. (email: amanda.dawson@griffithuni.edu.au)

Microplastics (plastics <5 mm) have been isolated from biota representing the full spectrum of feeding mechanisms, habitats, and trophic levels from zooplankton to megafauna[1]. Marine microplastics are attributed to two main sources; the direct release of micro-sized plastic particles into the environment and the in situ environmental breakdown of larger plastics. Microplastics are prevalent in the marine environment, and degradation occurs continuously on unknown timescales until the polymer is completely mineralised into carbon dioxide, water and biomass[2]. All microplastics are expected to continue fragmenting, thus reaching nano sizes (<1μm). Thus microplastics in the environment are heterogeneous in size and in shape[3], and consequently present a challenge for standardised monitoring[1].

Planktonic suspension and filter feeders may be the most susceptible to microplastic ingestion due to the relatively indiscriminate nature of this feeding strategy[4]. In particular, polyethylene (PE), polypropylene (PP), and expanded polystyrene (PS) are all less dense than seawater, making them buoyant and available to planktonic species[5]. Detrimental health effects have been associated with physical obstruction of the digestive system and associated reduced nutritional condition[6].

Laboratory-based feeding studies are a commonly used approach for the quantification of exposure and associated effects. Often these studies use invertebrate species such as zooplankton, which form the basis of the pelagic food web. Ingestion at this level therefore carries a threat of plastic bioaccumulation and biomagnification to higher trophic levels[1]. Organisms are exposed to relatively homogenous, commercially available, plastic beads to replicate environmental condition[3]. Such studies have confirmed numerous planktonic species are capable of ingesting and egesting microplastics[7–11], many of which were associated with toxic and physiological effects[2,12–15]. Despite a growing body of exposure and affect assessments, the ecological consequences of microplastic ingestion by zooplankton remain unclear. Further, the fate and degradation of microplastics, as a consequence of ingestion is rarely considered.

Here we expose Antarctic krill (*Euphausia superba*, hereafter 'krill'), a keystone species in the Antarctic ecosystem, to PE microbeads (27–32μm diameter) along with an algal food source to determine the fate of microplastics ingested by a planktonic crustacean of high dietary flexibility and ecological importance. Krill predominantly feed on phytoplankton but regularly prey on other zooplankters including salps, copepods and other krill[16]. In terms of biomass, Antarctic krill are extremely abundant, supporting a large number of Southern Ocean consumers[17,18] and are a major phytoplankton grazer in the Southern Ocean[18,19]. Krill filter feed by forming a feeding basket through which water is passed (Supplementary Fig. 1). Food particles are retained on the basket and are then transported to the mandibles for mastication[20,21]. The mandible, situated at the base of the oesophagus, is equipped with a cutting and grinding surface[22]. Food is then directed through the short oesophagus into the stomach and gastric mill where it is mixed with digestive enzymes for further mastication[23,24]. Thereafter, particles smaller than the primary filter (0.144 μm) pass through to the digestive gland, and larger particles are directed to the mid and hind gut for egestion[25]. Egested particles are encased in a peritrophic membrane which protects the mid and hind gut from abrasion[26].

The digestive gland is the primary site for cellular digestion (Supplementary Fig. 1). The gland is made up of groups of blind ending tubules, which are comprised of epithelial cells. Food particles that enter the digestive gland are pumped into the tubules, where digestive enzymes are directly released, thus allowing for nutrient adsorption and intracellular digestion to take place[26–28].

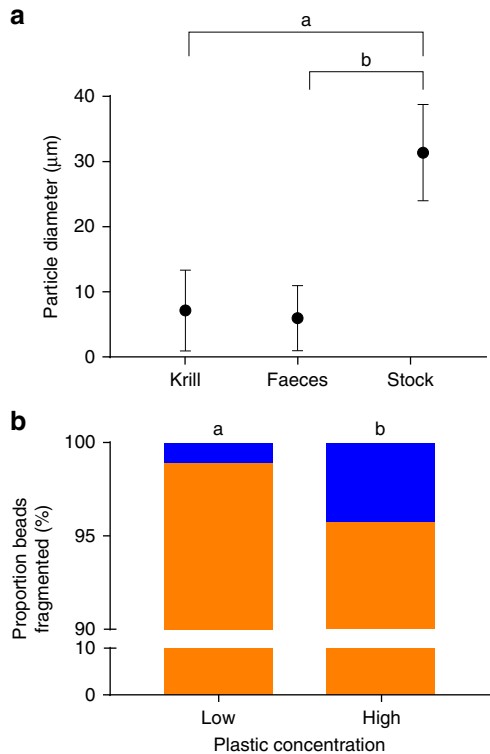

**Fig. 1** Polyethylene particle size and proportion fragmented. **a** Microplastic particle size (mean ± S.D) in all sample types: whole krill homogenates, egested faecal pellets, and in the exposure stock suspension, letters denote statistically significant differences ($p < 0.05$, two-sample Kolmogorov–Smirnov Test). **b** Proportion of whole beads (blue) and fragments (orange) isolated from Antarctic krill (*Euphausia superba*) exposed to a Low (20%) ($n = 9$ krill with 16,308 particles detected) and High (80%) ($n = 9$ with 51,168 particles detected) plastic concentrations. Letters denote statistically significant differences ($p < 0.05$, $X^2$ test)

We examine exposed krill and their faecal material microscopically to (1) quantify the size of particles present in the krill digestive system and in egested material, (2) identify where these particles are localised within the digestive system, and (3) examine the effect of particle size on egestion. We find that Antarctic krill are capable of fragmenting pristine PE microbeads into significantly smaller fragments, showing that nanoplastics can be generated by the ingestion of microplastics in a marine species.

## Results

**Antarctic krill fragment ingested virgin PE**. To determine the effect of ingestion on microplastic beads we exposed krill to a 4-day static renewal assay, which incorporated daily feeding on two PE microplastic and algal diets. A 'low' diet was comprised of 20% plastic and 80% algae; the 'high' diet was comprised of 80% plastic and 20% algae. Krill were exposed daily for 4 h to their diet; this was followed by 20 h in clean filtered seawater (Supplementary Fig. 2). Whole krill were enzyme-digested after exposure to isolate the ingested microplastics, as was faecal material collected throughout the experiment. We compared the size distribution of particles from the stock suspension to the distribution of particles within the krill and egested faecal pellets. We found all krill contained a mixture of whole PE microplastic beads and PE fragments that was not consistent with the exposure stock. Beads in the stock suspension had a mean diameter of 31.5 μm (±7.6 standard deviation, S.D), whereas the mean particle size

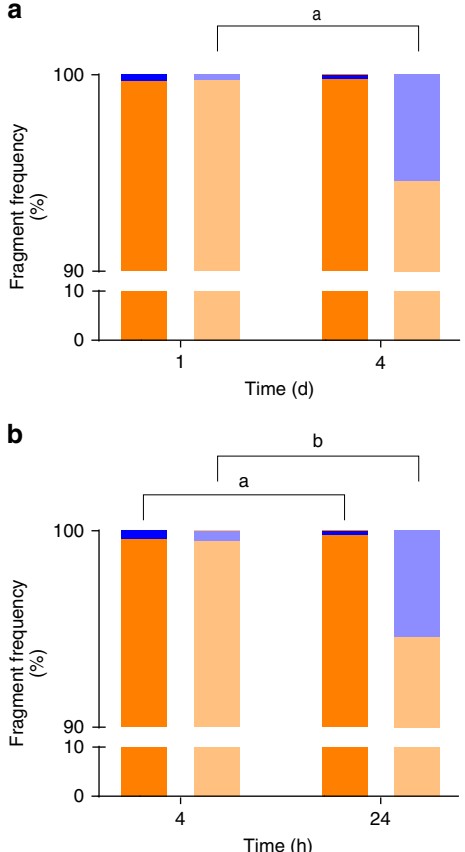

**Fig. 2** Proportions of egested fragmented and whole beads over time. Frequency of whole (blue) and fragmented (orange) particles isolated from faecal pellets of Antarctic krill exposed to Low (20%; $n = 3$ beakers) (dark) and high (80%; $n = 3$ beakers) (light) concentrations at: **a** 24 h on Day 1 ($n = 6$) and Day 4 ($n = 6$), **b** 4 ($n = 6$) and 24 h ($n = 6$) on Day 4 only. All faecal material per beaker (containing five krill) was pooled to form a single sample per time point per dose. Letters denote statistically significance differences in the proportion of whole beads excreted over time ($p < 0.05$, $X^2$ test)

isolated from within the krill was, on average, 78% smaller than the original beads ($7.1 \, \mu m \pm 6.2$ S.D), with some fragments reduced by 94% of their original diameter. Particles isolated from faecal material were also reduced ($6.0 \, \mu m \pm 5.0$ S.D). Further, the size distribution of particles within the krill, and excreted particles, were significantly different to beads in the exposure stock (two-sample Kolmogorov–Smirnov tests, $D = 112$, $p < 0.001$ and $D = 113$, $p < 0.001$, respectively) (Fig. 1, Supplementary Fig. 3). The reduced plastic particle size found in krill and their faecal pellets revealed that Antarctic krill were physically fragmenting beads after ingestion. We found no relationship between krill size and their ability to fragment plastics (multiple linear regression, $F_{3,15} = 2.595$, $p > 0.05$, $R^2 = 0.357$).

To ensure that the homogenisation process was not responsible for fragmenting the beads we carried out procedural blanks. These consisted of whole krill enzyme digested, beads enzyme digested and beads not subjected to any digestion or homogenisation. Beads were unaffected by the sample analysis procedures, neither the homogenisation process nor the digestion enzymes were responsible for fragmenting the beads.

Particles from the krill and bead blanks were found to be unaffected by the enzyme digestion protocol. Visually, beads from the stock suspension appeared similar to the bead blanks. As did the whole beads and fragments isolated from the krill and krill

blanks. The distributions of particle sizes from experimental and blank samples were very similar, despite unequal sample sizes (Supplementary Fig. 3). Overall it was determined that krill were responsible for fragmenting the beads.

**Repeated exposure decreases fragmentation**. Notably, not all ingested beads were fragmented in the current study. To further explore this observation we compared the proportion of fragments to whole beads isolated from whole krill homogenates and faecal pellets exposed to the high and low treatments. The proportion of fragmented beads egested by the krill on days 1 and 4 were compared to assess the effect of repeated exposure. An extra sample point was added on day 4 to assess fine scale temporal variation within a daily cycle after repeated exposure.

Whole beads were found in the stomach and midgut content, as well as faecal pellets. Exposure concentration played an important role in the ability of krill to fragment the PE beads; where lower plastic concentration appeared to facilitate the krill's capacity to triturate plastic. Krill contained significantly more whole beads when exposed to a high plastic diet than a low plastic diet ($X^2_1 = 323$, ($N = 67476$), $p < 0.001$) (Fig. 1). Faecal pellets also followed this trend ($X^2_1 = 600$, ($N = 54670$), $p < 0.001$). Further examination revealed a significant interaction between time, dose, and the proportion of fragmented plastic (two-way analysis of variance (ANOVA), $F_{1,45778} = 328$, $p < 0.001$). Increased dose and repeated exposure appeared to inhibit the ability of krill to triturate plastic. Faecal pellets of high-dose krill collected after the first day of exposure contained a lower proportion of whole beads than faecal pellets collected after the final day of exposure ($X^2_1 = 384$, ($N = 27317$), $p < 0.001$) (Fig. 2). Whereas, when comparing the first and last day of exposure, krill exposed to low dose plastic appear capable of fragmenting plastics irrespective of repeated expose ($X^2_1 = 2$, ($N = 18465$), $p > 0.05$) (Fig. 2). Faecal pellets of high dose krill collected at 4 and 24 h on the last day of exposure clearly show an increasing trend of whole beads being egested over the final 24 h ($X^2_1 = 238$, ($N = 24828$), $p < 0.001$) (Fig. 2). The low-dose and high-dose krill both exhibit similar proportions of egested whole beads at 4 h on the last day of exposure. However, where the high dose krill appear to decrease their ability to fragment plastics over time, the low-dose krill exhibited the opposite trend over the final 24 h. Krill exposed to the low dose egested a higher proportion of fragments suggesting more efficient fragmentation ($X^2_1 = 5$, ($N = 17175$), $p = 0.018$). Overall it appeared that krill at the beginning of each daily pulse exposure were efficient at fragmentation. As krill ingested more beads the fragmentation efficiency decreased.

**Tissue localisation of fragments**. To further investigate plastic fragment kinetics within the organism, histological cryosections of exposed krill were prepared. We observed microplastics within the oesophagus, stomach, digestive gland and midgut of deceased krill (Fig. 3, Supplementary Fig. 1). Plastic were also visible in the stomach of live krill. Mandibles frequently had plastic fragments enmeshed in the grinding surface. The bulk of plastic maceration presumably took place in the stomach and gastric mill, which is responsible for mechanically fragmenting food particles under normal feeding conditions. Due to their predominantly herbivorous diet, Antarctic krill have complex digestive enzymes with high activity[18]. In this study we did not examine the effects of digestive enzymes on microplastics thus cannot rule out the possibly that digestive enzymes contributed to the fragmentation displayed in this study. Small food items then pass through a filter (approximately $0.14 \, \mu m$) into the digestive gland. Thus, large

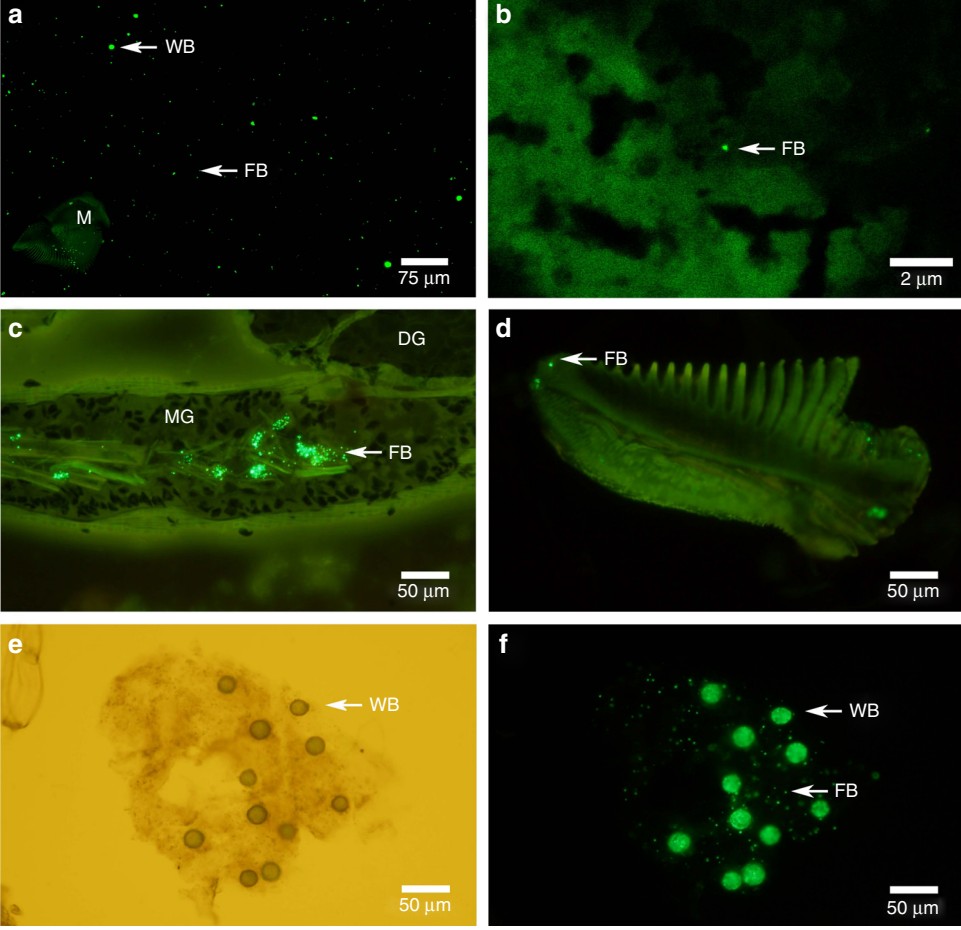

**Fig. 3** Fate of polyethylene beads and fragments after ingestion by Antarctic krill. Krill ($n = 17$) were used for histological analysis. **a** Beads on a filter paper isolated from digested krill with autofluorescent mandible, **b** Digestive gland tissue, **c** Midgut and digestive gland tissue, **d** Mandible with polyethylene fragments embedded in the surface, **e** and **f** Faecal pellet with polyethylene beads under bright field and fluorescence microscopy. WB: whole bead, FB: fragmented bead, M: mandible, DG: digestive gland, MG: midgut

plastic fragments and full sized beads were excluded from the digestive gland and directed to the midgut for excretion.

Microscopic limitations precluded a comprehensive investigation into the size and abundance of fragments found in the digestive gland. However, we detected particles in the digestive gland of two out of the five krill examined, within an approximate size range of 150–500 nm. The digestive gland is responsible for the absorption of digested material into the haemolymph[29]. The presence of PE fragments in the digestive gland revealed krill triturate PE beads to colloidal sizes, which increases the capacity for crossing biological barriers[30].

**Size-dependent egestion**. To examine egestion, we exposed krill to low-dose plastic for 10 days, after which their diet was swapped to 100% algae. Faecal pellets were collected for 5 days following the diet change. Small triturated fragments were more persistent and retained within the krill's body for longer than large beads. The proportion of whole beads excreted by krill decreased significantly throughout the egestion period ($X^2_4 = 16$, ($N = 21525$), $p = 0.003$), with whole beads no longer excreted after three days following the diet change (Fig. 4). Fragments were present in faecal material throughout all samples. This finding corresponds well with previous observations of size dependent egestion in marine invertebrates, both in laboratory and wild caught species[31–33].

## Discussion

Despite a growing body of research, there are still considerable knowledge gaps regarding spatial patterns and abundance of microplastics in the marine environment. The paucity of studies concerning microplastic ingestion in wild caught zooplankton hampers comparisons to this study. Microplastics isolated from euphausiids and other zooplankton in the wild have been found to range in size from 123 to ≤2000 µm[7,34], which is more than two orders of magnitude larger than the bead fragments Antarctic krill were found capable of producing in this study.

The phenomena of digestive fragmentation has never before been reported in other planktonic crustaceans, such as copepods or isopods, despite the fact that many possess similarly developed gastric mills and mouthparts designed for mechanical disruption[29]. However, copepods are theorised to scrape biofilms from the surface of pelagic plastics, inadvertently consuming liberated plastic fragments[35]. We hypothesise the absence of this observation in other planktonic crustaceans in the laboratory may be due to the use of different polymers in experiments. Two of the most commonly used laboratory plastics for feeding studies, PE and PS, differ in mechanical properties. The more commonly used PS is a rigid plastic, with a higher capacity to withstand stress than PE[36].

Regardless of their original polymer properties, marine microplastics are largely comprised of secondary plastics, derived from the breakdown of larger plastic items[3,37]. These secondary

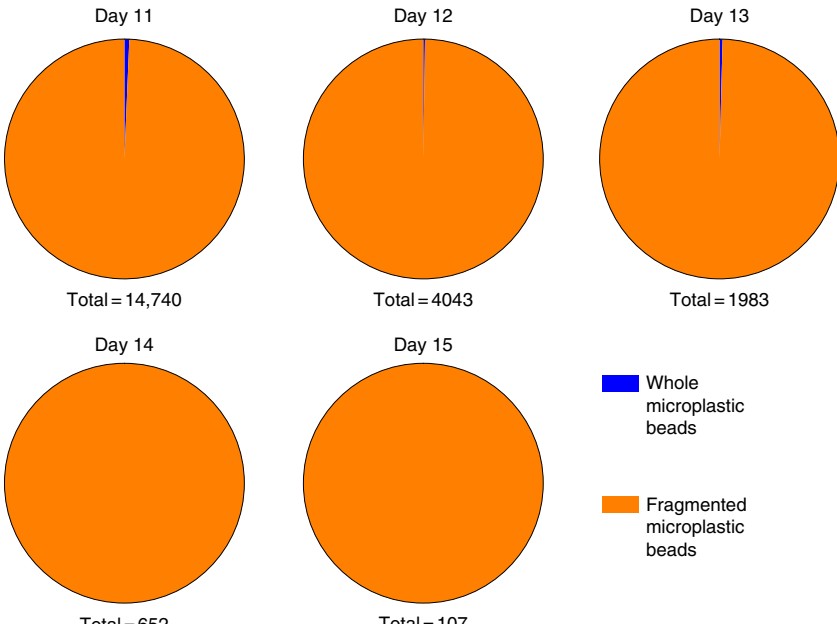

**Fig. 4** The proportion of PE plastic fragments to whole beads isolated from Antarctic krill. Faecal material (from $n = 15$ krill) was collected over 5 days, after switching from 10 days of low dose microplastic exposure, with daily static renewal, to an uncontaminated algae diet. Total refers to the total number of particles measured in each 24 h period of faecal material. Fragments are shown in orange, while whole beads are shown in blue

plastics are subject to weathering and chemical degradation rendering them physically and chemically altered from virgin plastics, such as those used in this study. Weathering serves to reduce the mechanical strength of plastics, leaving them brittle[38,39]. While the capacity of zooplankton to fragment secondary plastics requires further study, we suggest that embrittlement of secondary plastics will facilitate digestive fragmentation. We hypothesise fragmentation of microplastics after ingestion may be more common in the environment than the published literatures currently demonstrate. Previous observations of crabs altering laboratory-degraded fibres after ingestion offer weight to this hypothesis[40].

Nonetheless without further testing on other polymers and microplastic particles with varying degrees of degradation, it is difficult to speculate the frequency at which microplastic fragmentation in the environment could be occurring. PE is one of the most common plastics in the marine environment[41,42], thus even if this phenomenon is restricted wholly to PE; it still could present a significant pathway of microplastic degradation in the marine environment.

In general, PE has a low resistance to ultraviolet (UV) degradation and recent studies have identified that PE microplastics collected from the North Atlantic subtropical gyre were considerably weathered, with shorter polymer chain lengths, reduced molar mass and were more crystalline than reference PE[39]. Glassy polymers such as PS or PE terephthalate, however, are stronger and less susceptible to UV degradation[38]. Despite the properties of pristine polymers, all plastics, even those with chemical stabilisers, will eventually degrade in the environment.

The low-exposure concentration applied in this study was within the same order of magnitude as microplastic concentrations observed in pelagic systems of the North Pacific Subtropical Gyre[2], which are among the highest concentrations reported globally. Limited pelagic microplastics surveys from the Southern Ocean isolated between 0.0032 and 1.18 particles m$^{-3}$ [43,44], these levels are considerably less than those used in this experiment. In spite of the elevated exposures used in these experiments, considerable bead fragmentation was achieved. These preliminary

findings, although limited by scarce environmental data related to plastic <330 μm in natural marine systems, suggest that current concentrations may be within the bounds of optimal trituration for krill, but fragmentation efficiency may be affected by chronic exposure. The increased fragmentation of plastic noted at low-exposure conditions gives further weight to our hypothesis that digestive fragmentation is more common in the environment than recorded in current literature, which often use similarly high exposure concentrations for exposure experiments[3]. Current contamination levels in the Southern Ocean are theoretically low enough to promote efficient digestive fragmentation by krill species, and in a global context, possibly for other zooplankton with sufficiently developed gastric mills.

We did not examine these fragmented particles for induced toxicological effects. Several laboratory studies have demonstrated the ability of micro and nanoplastics to translocate to the haemolymph[45–47], however in these studies, the exposure particle size was sufficiently small to achieve translocation. We identify the potential for translocation to occur after an organism has physically altered the ingested plastics. This reveals a previously unidentified dynamic in the plastic pollution threat, with the implication that biological fragmentation of microplastics to nanoplastics may be widespread within ecosystems. As such, the harmful effects of plastic pollution must take into consideration not only the physical effects to the individual of macro and microplastic ingestion, but also the potential cellular effects of nanoplastics and the ecosystem impacts of biomagnification hereof. The effect of nanoplastics on crustaceans is unknown, although previous studies observed PE microbeads to induce genotoxicity and immunological effects in haemocytes[47].

The study relied on a combination of microscopy and quantitative image analysis of fluorescence intensity and size of plastic beads in krill faecal material and digested tissue, which was compared to a suspension of beads that were not exposed to krill. A rigorous experimental design for microscopic imaging and analysis was applied and we are confident, in spite of only using a fluorescence-based methodology to measure the beads, that this study gives evidence of krill degradation of ingested plastics. The

strong fluorescent signal present in all exposed krill was not present in the control krill. Controls were not exposed to the microplastics beads, but had been exposed to algal food and were enzyme digested in the same method as the experimental krill. Furthermore, the trends of fragments and whole beads were dose-dependent. This demonstrates that the fluorescence was not derived from background biological material, i.e., undigested algae, or exoskeleton. Three different procedural blanks, whole unhomogenised microplastic-fed krill, whole unhomogenised beads enzyme digested, and whole unhomogenised beads without enzyme digestion, were used to confirm that fragmentation was not due to the enzyme digestion and that enzyme digestion did not affect the detectability of the beads. The images were generated using identical microscopy conditions between controls and treatment samples, and the use of an automated macro for analysis ensured that the data were not at risk of subjective analysis.

Previous studies have suggested relatively simplistic interactions between microplastics and zooplankton[7,9,10] and biota-facilitated plastic degradation (considered to be predominantly undertaken by microorganisms) is currently considered negligible in the marine environment[2]. However, our results bring into question these previous conclusions. The fate of these altered particles, after egestion, death or predation is completely unknown, and is not necessarily comparable to non-ingested particles. Studies that neglect these interactions may be neglecting a significant pathway of degradation. Interestingly, ter Halle, et al.[48] recently showed that smaller microplastics are fragmented faster than larger particles under environmental conditions. The repercussions of organisms accelerating this process deserve further study.

It is also possible that fragmentation resulted from, or was enhanced by, the presence of silica diatoms in the diet. The churning and grinding action of the gastric mill combined with the sharp edges of triturated algae may have fragmented the beads. This could explain the decreased fragmentation in the high-exposure treatments, where there was a correspondingly lower algal concentration. However, this mechanism does not explain the temporal variation in fragmentation efficacy with repeated exposure, as krill diet within treatments remained constant over time. Thus fragmentation may have been enhanced by the presence of silica diatoms, but it was unlikely to be the sole cause of fragmentation.

This study uncovered the ability of an Antarctic keystone species to physically change ingested microplastics in a manner not previously described and in doing so, provides evidence for biologically facilitated production of nanoplastics. We hypothesise fragmented microplastics have increased potential for interaction at the molecular level, as seen in other nanoplastic studies[12], and this warrants significant attention to nanoparticle toxicology in the discussions surrounding global plastic pollution. Triturated microplastics will likely impact potential particle bioavailability and biomagnification, and likely influence the timescales needed for complete mineralisation.

## Methods

**Microplastic characterisation**. A microplastic feeding stock suspension was made from commercially available (Cospheric LLC CA, USA—UVPMS-BG-1.025) fluorescent green PE microbeads (27–32-µm diameter, 1.030 or 1.026 g cm$^{-3}$). The beads were confirmed to be PE by Fourier transform infrared (FTIR) spectroscopy using a PerkinElmer FTIR spectrometer (Supplementary Fig. 4). The bead size range was selected to closely conform to the size range of the algal food, simultaneously offered to the krill (see below). Density was selected to be close to neutrally buoyant in 0 °C seawater. The physical properties of the microbeads were characterised using images of beads subsampled from the feeding stock (see Sample analysis section below).

**Exposure design**. Mixed sex Antarctic krill were collected from the Southern Ocean (66.33 S, 59.34 E) in the Austral summer of 2014/2015. Krill were maintained in the Marine Research Facilities at the Australian Antarctic Division, Tasmania according to previously established methods until use in experiments[49]. The exposure design followed previously described methods[50,51]. Adult krill ($n =$ 65, wet weight: 0.556 ± 0.117 mg, length: 41.1 ± 3.7 mm) were acclimatised for 24 h prior to the start of experiments in 5-L glass beakers. Krill were randomly selected for use in the experiment from apparently healthy free swimming schooling adults. Krill were collected into buckets by repeatedly dipping a small net into the same region of the tank as the krill schooled anticlockwise. Buckets contained 15 krill; these were randomly distributed amongst beakers, so each beaker contained five adult krill in 4 L seawater. Five krill per 4 L is the maximum density krill can be maintained under experimental conditions. Block randomisation was applied to distribute krill amongst treatments, and the investigator was not blinded to the treatments. The sex of individuals was not determined in the experiment. Seawater temperatures were maintained at 0 °C (±0.5) and beakers were kept in total darkness throughout the experiment but were exposed to a small amount of red light from a headlamp worn by handlers during the water changes. Exposure seawater was collected from Bruny Island, Tasmania, and filtered to 0.2 µm. Filtered seawater was pre-chilled to 0 °C (±0.5) before krill were added. The dietary exposure suspension was prepared daily from stock using fluorescent plastic microbeads with concentrated instant non-viable algae *Thalassiosira weissflogii* (Reed Mariculture, CA, USA). The size range for *T. weissflogii* cells was 5–20 µm according to the manufacturer. Although this is slightly smaller than the microplastics beads, Antarctic krill can feed efficiently on particles >2 µm up to whole zooplankton (~3 mm). Dietary exposure suspensions were made up as a portion of the krill's dietary requirements under laboratory conditions, 100% algae equates to 0.00798 mg *T. weissflogii* (dry weight) per beaker. Harvested krill were euthanised in liquid nitrogen or formalin. The seawater physiochemical parameters for the two experiments are outlined in Supplementary Tables 1–2. Mortality for all experiments is given in Supplementary Note 1.

**Particle size experiment**. Four-day feeding and egestion experiments were carried out on 45 Antarctic krill. Nominal daily exposure suspensions were made up to 20 or 80% microplastics by weight, which equated to approximately 29 or 116 beads mL$^{-1}$ (402 or 1606 µg L$^{-1}$). Control krill were fed 100% algae and all treatments were carried out in triplicate. Krill were transferred daily to exposure suspension and allowed to feed for 4 h, before being transferred with a stainless steel dip net to a clean beaker for 20 h. Before transfer, krill were flushed with cold fresh filtered seawater to remove plastics that may adhere to the exoskeleton. Upon transfer to the exposure suspension, krill were observed to be feeding almost immediately. Faecal pellets were collected after 24 h exposure on days 1 and 4. An extra sample point was added on day 4 to assess fine scale temporal variation after repeated exposure, thus faecal pellets were collected at 4 and 24 h on day 4 (refer to Supplementary Fig. 2). All beakers were harvested for particle size and tissue localisation analysis after 96 h. Three krill from each beaker were randomly selected for particle size analysis ($n = 18$ krill total). As the beads were fragmented after ingestion, the total bead ingestion rates could not be calculated from stomach content or egested material.

**Tissue localisation experiment**. To investigate tissue localisation of ingested plastic, two krill from each beaker were randomly selected, fixed in formalin, and used for histological cryo-section (20 µm) analysis. Slides were stained with H&E or remained unstained. Slides were examined using an Olympus BX60 fluorescence microscope or Zeiss-780 Laser Scanning Confocal microscope with a fluorescent filter of 488-nm excitation and 526-nm emission.

In addition, to investigate if krill could fragment plastics <1 µm and the possibility of fragments entering the digestive gland, five krill were exposed to 100% plastic diet (approx. 2063 µg L$^{-1}$ or 149 beads mL$^{-1}$) for 24 h, with no water changes. These five krill were all used for tissue localisation analysis.

**Egestion experiment**. To examine particle sizes egested over an extended period, 15 krill divided into three beakers were exposed for 10 days to a 20% diet (approx. 401 µg L$^{-1}$) of plastic following the same basic design as the Particle Size Experiment. After 10 days, the diet was switched to 100% algae for five days. Faecal pellets were collected at 4 and 24 h every day of the five day egestion period. Faecal material was pooled per beaker per 24 h resulting in 15 samples.

**Sample analysis**. Body burden analysis was carried out using an enzyme digestion followed by visual identification of ingested microplastics under a fluorescent microscope. Krill were flushed with Milli-Q water, blotted dry, weighed (to 3 d.p.), and heated to 65 °C in a water bath, after which the exoskeleton was removed. Krill were then homogenised using a glass rod, and digested using proteinase K adapted from Cole et al.[52], which was previously shown to have negligible effects on PS bead integrity. Digestion efficacy was not optimal as hard chitinous structures often remained after digestion. Digested krill were filtered under vacuum onto Millipore gridded 0.45 µm filters and air dried overnight. Filters were fixed between glass coverslips and analysed for microplastics using a Zeiss-780 Laser Scanning Confocal microscope with a fluorescent filter with a Plan-Apochromat 10 × /0.45 M27

lens, with a numerical aperture of 0.45. Microplastic fragments were imaged in five randomly selected squares ($6.97 \times 6.98$ mm; total area of 2.4 cm$^2$) on the filter paper, which accounted for 25% of the total filtered area. Images were verified by eye, and compared to controls to examine for undigested chitinous material with autofluorescence. Of the 165 images taken, 2 images were excluded on the basis of chitinous material with autofluorescence (See Fig. 2, panel A for example of excluded image). These were too large to be mistaken as a microplastic beads and were clearly distinguishable as mandibles. The diameter (major axis when particles were fitted to an ellipse) of each particle within each image was measured using imaging software (FIJI GPL v2)[53]. A minimum threshold was applied to the fluorescence intensity of each image to ensure only beads were counted by the imaging software. Thresholds were set to a minimum of 65 and maximum of 255 which allowed background material, including undigested exoskeleton (except for mandibles), algal cells and the filter paper, to be excluded without interference to the analysis. Size exclusions were applied to particles which had a diameter >50 μm, on the basis these were two or more beads too close together for the imaging software to distinguish individual beads and accurately measure size. See Supplementary Note 1 and Supplementary Figs. 5–6 for further information.

**Bead fragmentation**. To test that the sample analysis procedures were not responsible for fragmenting the beads, procedural blanks were carried out in a pilot study and throughout the experiment. Procedural blanks consisted of krill and beads or just beads. Krill blanks consisted of seven krill taken from a pilot study. The krill were digested as per the method described in the Sample Analysis section, except the krill were not homogenised. After digestion, exoskeleton remained intact but the tissue was completely digested, krill were vortexed and the stomach was opened to liberate any remaining beads. The sample was then filtered and imaged as per the method described in the Sample Analysis section. Bead blanks consisted of beads in the absence of krill and were not homogenised. Beads were added to buffer and enzyme, then digested, filtered and imaged as per methods outlined in the Sample Analysis section. Bead blanks were examined after enzyme digestion with FTIR spectroscopy (Supplementary Fig. 7), but ingested beads and fragments were unable to be detected on the cellulose filters with FTIR due to the low concentration and/or small size of the particle.

**Statistical analysis**. Two-sample Kolmogorov–Smirnov tests (two-tailed, $\alpha \leq 0.05$) were used to compare the particle size distribution from the stock microbeads to the size distribution of plastics isolated from the digested krill, and from the particles isolated from the faecal pellets. The proportion of whole beads compared to fragments in digested krill and in faecal pellets was compared between doses using Chi-squared analysis (two tailed, $\alpha \leq 0.05$). For all proportion tests (Chi-squared and linear regression), beads with a diameter $\geq 25$ μm were classified as whole beads, beads <25 μm were considered fragments. This cut off was selected by eye using the standard distribution of the stock beads. Kolmogorov–Smirnov tests (two tailed, $\alpha \leq 0.05$) were used to test for normality. The data were log$_{10}$ transformed and comparison between fragment size, sample time and plastic dose in the faecal pellets was determined with a two-way ANOVA (two tailed, $\alpha \leq 0.05$). Multiple linear regression was used to examine relationships between the length and weight of the krill and their ability to fragment plastics. Means are expressed as mean ± standard deviation (S.D.) unless otherwise stated.

**Data availability**. Data and image macro coding are available from the corresponding author on request.

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

## Acknowledgements

A.D. was supported by PHD scholarship from SOPOPP. We thank S. Wild and P. Eisenmann. for assistance with laboratory work, and staff at the Australian Antarctic Division for maintenance of krill in the Marine Research Facilities. We thank M. Arthur and C. Wild for providing statistical advice. N. Waterhouse for assisting design of imaging methods, T. Nguyen for imaging samples and staff at the QIMR Berghofer Medical Research Institute for sectioning the krill.

## Author contributions

A.D. and S.B.N. conceived of the idea of this study and provided financial means. S.K. and K.T. provided significant input on experimental design. A.D. and C.K. preformed laboratory experiments. R.K. and A.D. interpreted histological data. W.H. and A.D. designed image analysis methods. A.D. analysed the data and prepared the manuscript, all authors contributed substantially to revisions.

## Additional information

**Competing interests:** The authors declare no competing interests.

