## [Peer Review File · Nature Communications]

Reviewers' comments:

Reviewer #1 (Remarks to the Author):

This study shows for the first time that ocean microplastics may be broken down by the digestive system of zooplankton. This is a novel finding, with relevance for our understanding of ocean plastics' fate and impacts.

Using a set of feeding experiments with Antarctic krill, the authors show that ingested plastics can be broken down by the digestive system of krill, with some of the created fragments being small enough to (1) reach the digestive gland and (2) stay longer inside the krill's body.

My main recommendation is for the authors to substantially improve the manuscript structure and grammar. Otherwise this is not suitable for publication in a high impact journal such as Nature Communications.

Bellow I suggest a few revisions.

Abstract

Line 27: It may be good to give your reader a brief description of what you mean by microplastics/nanoplastics/microbeads in your abstract. I also think you don't need to use these 3 words in abstract. Using only 1-2 of them would decrease the number of definitions you need to provide here.

Line 30: Suggest replacing 'pristine microbeads' by 'microplastics'.

Line 31: Perhaps replace 'microplastics' by 'particles' and add the size range of your experimental particles in between brackets. Also, if you are not sure that the degradation process is 100% mechanical, I suggest rephrasing 'were mechanically fragmented to nanoplastics' into 'were fragmented into pieces (? - ? nm in length)'.

Line 31: I suggest replacing 'microbeads' with 'microplastics' throughout your abstract. Microbeads are a type of microplastics and your inferences do encompass secondary microplastics too.

Line 33: add 'into sizes' between words 'fragmented' and 'small'.

Line 36: "(...) but also introduces (...)". Suggest re-wording this into a new sentence.

Introduction

Line 44: Delete 'possibly smaller' as your nanoplastic definition included all particles <1micron.

Line 49: Add word 'expanded' before 'polystyrene'?

Line 63: Suggest replacing 'microplastic polyethylene beads' by 'polyethylene microbeads'. I also suggest adding the size range between brackets here as I think size range is a very important parameter of your experiment and the reader may want to know this early on during reading.

Lines 65 – 69: Suggest deleting the methods info going from "The krill (...)" to "(...) microscopically". I think you don't need to explain this here to state your main finding (lines 69 – 70). Furthermore, there are other experiments that you did not describe in here so this is incomplete.

Results

Many sentences in your results actually belong to the Discussion section. I suggest making sure to transfer all inferences to the Discussion section, or pulling both findings descriptions and inferences into one manuscript section entitled 'Results and Discussion'.

Lines 74 – 75: This sentence requires revision. It is not really clear what you mean here. Perhaps re-word and transfer the overall methods info you provided in the last paragraph of introduction to here and then explain what was compared (size distribution plastic particles in stock solution with those within krills, and faecal pellets?).

Line 79: Here you state particles' average size went from 31.5micron (stock solution) to 7.2micron (inside krill). Please also add what was the average particle size in the egested material.

Line 83: So you observed a decrease in the size of the microbead particles. How do you then conclude that this is a mechanical process, rather than a chemical+mechanical (e.g. digestive enzymes)? It may be interesting to dedicate a paragraph in your introduction for explaining the feeding ecology of krill. E.g. is the breakdown of diatoms done mechanically? I also think you should leave your inference on how this happens (mechanically, chemically+mechanically) to discussion. Results section is for description of what you observed, not inferences.

Lines 84 – 102: This type of info belongs to Discussion. I would say the putative scrape marks observed by Reisser et al. may have not been observed in experiments yet because ocean microplastics are far much more brittle than raw microbeads (independently of polymer type). One way to degrade particles for experiments would be to expose the microbeads to UV light prior to the experiment (similarly to what polymer biodegradation researchers do).

Line 130: "The fragmentation of ingested plastics may be related to krill gut capacity" This sentence belongs to Discussion.

Lines 135 – 149: In my view, this is also Discussion.

Lines 160 -173 and lines 180 - 182: Once again, Discussion material?

Figure 1

In your panel B, I suggest having your y-axis in % so the proportions of fragments become more visible. I also suggest adding more information to your legend so a reader can understand the figure with no need to refer to the text (e.g. what you mean by 'Krill', 'Faeces', 'Stock' in panel A, etc).

Figure 3

I found it quite hard to visualize patterns in this figure. Do the particle size differences between different days and times become clearer if you place y-axis in log scale?

Methods

Lines 319 – 320: So some of your biological material have autofluorescence. How do you know that the small fluorescent particles found within the krill and faecal pellets are not biological material instead of microbead fragments?

Lines 339 – 341: I find this sentence far better than the one in lines 74 – 75. I think they are meant to say the same?

Line 348: time and dose of what? Please specify.

Supplementary Text

First paragraph of 'Microplastic Characterization': So here you explain how you made sure the fragments considered were not just colorant flakes. Is it possible to also add info on how you made sure these fragments were not of biological origin?

Fig. S7: Suggest adding arrows to indicate where the microbead fragments are in these images.

Reviewer #2 (Remarks to the Author):

The ms. reports extremely interesting data on fragmentation of commercial microplastic in Antarctic krill. After exposure to different concentrations of microplastic, the size of the particles was analyzed in whole krill homogenates and fecal pellets at different times post exposure.

In this light, the work is original and of great interest. However, the quality of the ms. is rather low in terms of ms. organization, clarity of experimental design and presentation of the results.

The main flaw of the work is that fragmentation is apparently determined only on the basis by microscopical observation of particle micro and nano fluorescent particles. However, the presentation and discussion of the results relies on precise quantitative data. Therefore, due to the importance of the results, the possible limitations of this approach should be considered.

1) How can be the Authors sure that all the particles measured were plastic items? I can understand that FTIR could not work on the samples, but counting and measuring nano and micro-sized green fluorescent dots in a marine invertebrate fed on algae is a difficult task.

2) How many samples for each type of determination were utilized? In the figures or text there is no mention of the number of samples on which statistical analysis was performed. For image analysis (Fig.2) apparently only 2 krill were utilized. How many for the homogenates and fecal pellets? These data must be clearly indicated.

3) In the methods section, what does 'Particle Size Bioassay' mean?

This weakness is apparently reflected in Abstract and introduction (last para), where great emphasis is given on the results obtained without clearly explaining HOW they were obtained.

Results:

This section is not well organized and therefore not clear. Results and discussion are mixed up in almost all the subsections. The sub-heading should be revised trying to group the main results obtained in whole homogenates and fecal pellets.

please refer to plastic 'suspension' and not 'solution' throughout the ms...

What is more, in this section it should be more clearly explained on how many samples measurements

were performed and how the size was determined, if not within the text at least in figure legends. Instead of discussing the interpretation of the results, the way data were obtained should be clearly indicated.

Discussion

The discussion should contain a critical and comparative evaluation of the results reported, before discussing their implications and significance .

A critical discussion of the methods used and on the reliability of the results obtained is needed.

Quantitative data must be compared with other studies reporting data on microplastic ingestion in terms of body burden and elimination with fecal pellets in other planktonic species.

The choice of the Antarctic species should be justified.

Moreover, since the concentrations of microplastic utilized were extremely high, and, fortunately, not in a predictable environmental range, the results should be also discussed in terms of the possible occurrence of fragmentation of microplastic by krill in real environmental conditions at much lower particle concentrations.

Reviewer #3 (Remarks to the Author):

This is an interesting study and well presented, I believe it to be worthy of publication. The major claim of the paper is that ingestion of microplastic by krill results in fragmentation of the plastic to smaller nanoplastics. The study suggests that exposure to a high concentration of beads or repeated exposure decreases fragmentation. These findings are novel and I believe will be of interest to many researchers in the field of microplastic research and zooplankton ecology. What is emerging from the growing literature describing microplastic abundance is the mismatch between how much plastic is entering the oceans, and how much is found floating on the ocean surface. Models have conservatively estimated that over 5 trillion pieces of plastic are floating in the world's oceans and yet this accounts for approximately 1% of the estimated ocean plastic load. This raises the key question; where is the remaining 99% of plastic going; which this study begins to answer?

I believe the conclusions of the paper are original and will influence thinking in the field of microplastic research. This study will contribute to our understanding the fate of microplastics in the marine environment and their potential impact on marine biota. A weakness of the paper is that the experiments have only been run with one polymer type (Polyethylene), the conclusions of the paper would be considerably strengthened if different polymer plastics as well as weathered and virgin plastics were used, however, I accept this is outside the scope of this study.

In summary I think the manuscript is worthy of publication following minor revision, see marked manuscript. The study is reproducible from the methods given and will make a good contribution to this topical and important growing area of research.

Dr P K Lindeque

	Reviewers' comments:	Author Response
	Reviewer #1 (Remarks to the Author):	
1	This study shows for the first time that ocean microplastics may be broken down by the digestive system of zooplankton. This is a novel finding, with relevance for our understanding of ocean plastics' fate and impacts. Using a set of feeding experiments with Antarctic krill, the authors show that ingested plastics can be broken down by the digestive system of krill, with some of the created fragments being small enough to (1) reach the digestive gland and (2) stay longer inside the krill's body. My main recommendation is for the authors to substantially improve the manuscript structure and grammar. Otherwise this is not suitable for publication in a high impact journal such as Nature Communications. Bellow I suggest a few revisions.	We thank the reviewer for their helpful comments on our manuscript. We have attempted to enhance the quality of the manuscript by significantly improving the structure and grammar throughout. See individual comments below for sections that have been restructured
2	Abstract	
3	Line 27: It may be good to give your reader a brief description of what you mean by microplastics/nanoplastics/microbeads in your abstract. I also think you don't need to use these 3 words in abstract. Using only 1-2 of them would decrease the number of definitions you need to provide here.	Line 28: Changed to "Microplastics (plastics <5mm diameter)"
4	Line 30: Suggest replacing 'pristine microbeads' by 'microplastics'.	Line 30: Changed to "microplastics"
5	Line 31: Perhaps replace 'microplastics' by 'particles' and add the size range of your experimental particles in between brackets. Also, if you are not sure that the degradation process is 100% mechanical, I suggest rephrasing 'were mechanically fragmented to nanoplastics' into 'were fragmented into pieces (? - ? nm in length)'.	Line 32: Changed to "Ingested microplastics (31.5µm) were fragmented into pieces (<1µm diameter)"
6	Line 31: I suggest replacing 'microbeads' with 'microplastics' throughout your abstract. Microbeads are a type of microplastics and your inferences do encompass secondary microplastics too.	The term microbeads has been changed to microplastics throughout the abstract
7	Line 33: add 'into sizes' between words 'fragmented' and 'small'.	Line 35: 'Into sizes' has been inserted
8	Line 36: "(...) but also introduces (...)". Suggest re-wording this into a new sentence.	Not adopted due to the strict word limits for the abstract. We feel this sentence accurately conveys its message without appearing overly verbose
	Introduction	
9	Line 44: Delete 'possibly smaller' as your nanoplastic definition included all particles <1micron.	'Possibly smaller' has been removed
10	Line 49: Add word 'expanded' before	Line 51: Changed to 'expanded

	'polystyrene'?	polystyrene'
11	Line 63: Suggest replacing 'microplastic polyethylene beads' by 'polyethylene microbeads'. I also suggest adding the size range between brackets here as I think size range is a very important parameter of your experiment and the reader may want to know this early on during reading.	Line65: Changed to "polyethylene microbeads (27 -32µm diameter)"
12	Lines 65 – 69: Suggest deleting the methods info going from "The krill (...)" to "(...) microscopically". I think you don't need to explain this here to state your main finding (lines 69 – 70). Furthermore, there are other experiments that you did not describe in here so this is incomplete.	Line 82-87: The section from 'the krill' - 'examined microscopically' has been removed. Instead we have used this section to clarify the aims
	Results	
13	Many sentences in your results actually belong to the Discussion section. I suggest making sure to transfer all inferences to the Discussion section, or pulling both findings descriptions and inferences into one manuscript section entitled 'Results and Discussion'.	After consulting the guide to authors and discussion with the Editor the authors have been advised that some interspersed discussion points are allowed in the results section. Therefore I have adopted many of Reviewer 1's recommendations but some small discussion points have been left in the results section. We feel these comments enhance the findings in the results section. Please see below comments responses for exact sections that have been moved to the Discussion
14	Lines 74 – 75: This sentence requires revision. It is not really clear what you mean here. Perhaps re-word and transfer the overall methods info you provided in the last paragraph of introduction to here and then explain what was compared (size distribution plastic particles in stock solution with those within krills, and faecal pellets?).	This section has been reworded to clarify. Line 90: 'To determine the effect of ingestion on the microplastic beads we exposed krill to a 4 day static renewal assay, which incorporated daily feeding on two (low - 20% and high - 80%) PE microplastic and algal diets. Krill were exposed daily for 4 hours to their diet; this was followed by 20 hours in clean seawater. Whole krill were enzyme digested after exposure to isolate the ingested microplastics, as was faecal material collected throughout the experiment. We compared the size distribution of particles within krill and egested faecal pellets with beads from the stock solution'
15	Line 79: Here you state particles' average size went from 31.5micron (stock solution) to 7.2micron (inside krill). Please also add what was the average particle size in the egested material.	Line100: Added: 'Particles isolated from faecal material were also reduced (6.0 µm ± 5.0 S.D).'
16	Line 83: So you observed a decrease in the size of the microbead particles. How do you then conclude that this is a mechanical process, rather than a chemical+mechanical (e.g. digestive enzymes)? It may be interesting to	Reference to mechanical fragmentation of plastics have been removed Line 67-70: Paragraph added to explain krill feeding ecology

	dedicate a paragraph in your introduction for explaining the feeding ecology of krill. E.g. is the breakdown of diatoms done mechanically? I also think you should leave your inference on how this happens (mechanically, chemically+mechanically) to discussion. Results section is for description of what you observed, not inferences.	Line 70-81: Paragraph added to explain role of stomach in normal phytoplankton digestion and the role of digestive enzymes
17	Lines 84 – 102: This type of info belongs to Discussion. I would say the putative scrape marks observed by Reisser et al. may have not been observed in experiments yet because ocean microplastics are far much more brittle than raw microbeads (independently of polymer type). One way to degrade particles for experiments would be to expose the microbeads to UV light prior to the experiment (similarly to what polymer biodegradation researchers do).	Line 178 – 185 This section has been moved to the Discussion The Reviewers suggestion of further experimentation with UV degraded particles is very interesting and will be considered for further experiments but was beyond the scope of this experiment. Whilst UV degradation, in addition to other abiotic process, may alter the rate of degradation by krill, it seems unlikely that any outcomes would change the premise of this study overall: that for the first time krill (and potentially other species) are noted to be part of the degradation process of plastic in the ocean.
18	Line 130: “The fragmentation of ingested plastics may be related to krill gut capacity” This sentence belongs to Discussion.	This sentence was removed
19	Lines 135 – 149: In my view, this is also Discussion.	Line207-220: This section was moved to the discussion
20	Lines 160 -173 and lines 180 - 182: Once again, Discussion material?	Lines 220-321: This section was moved to the discussion
21	Figure 1 In your panel B, I suggest having your y-axis in % so the proportions of fragments become more visible. I also suggest adding more information to your legend so a reader can understand the figure with no need to refer to the text (e.g. what you mean by ‘Krill’, ‘Faeces’, ‘Stock’ in panel A, etc).	Adopted: In Figure 1 Panel B colour has also been added to make the proportions more visible Fig 1 caption added ‘ Microplastic particle size (mean \pm S.D) in all sample types: whole krill homogenates, egested faecal pellets, and in the exposure stock suspension,
22	Figure 3 I found it quite hard to visualize patterns in this figure. Do the particle size differences between different days and times become clearer if you place y-axis in log scale?	Figure 3 changed to a stacked column graph to display the trends more clearly
	Methods	
23	Lines 319 – 320: So some of your biological material have autofluorescence. How do you know that the small fluorescent particles found within the krill and faecal pellets are not biological material instead of microbead fragments?	The following sentences have been added in the Sample Analysis section. Line 379: ‘These were too large to be mistaken as a microplastic beads and were clearly distinguishable as mandibles.’ And

		Line 381-385: 'A minimum threshold was applied to the fluorescence intensity of each image to ensure only beads were counted by the imaging software. Thresholds were set to a minimum of 65 and maximum of 255 which allowed background material, including undigested exoskeleton (except for mandibles), algal cells and the filter paper, to be excluded without interference to the analysis.' Further, control krill were analysed in the same way as experimental krill thus auto fluorescence was accountable by comparing these images.
24	Lines 339 – 341: I find this sentence far better than the one in lines 74 – 75. I think they are meant to say the same?	See above response to Comment 14. This sentence has been changed
25	Line 348: time and dose of what? Please specify.	Line 409: Added '...comparison between fragment size, sample time and plastic dose in the faecal pellets was determined...'
	Supplementary text	
26	First paragraph of 'Microplastic Characterization': So here you explain how you made sure the fragments considered were not just colorant flakes. Is it possible to also add info on how you made sure these fragments were not of biological origin?	See response above to comment 23. Further, non-homogenised krill were used as krill blanks. These samples did not contain autofluorescent biological material but did contain fragments in similar size ranges to the experimental krill.
26	Fig. S7: Suggest adding arrows to indicate where the microbead fragments are in these images.	Arrows have been added to this image to indicate the fragments
	Reviewer #2 (Remarks to the Author):	
27	The ms. reports extremely interesting data on fragmentation of commercial microplastic in Anctartic krill. After exposure to different concentrations of microplastic, the size of the particles was analyzed in whole krill homogenates and fecal pellets at different times post exposure. In this light, the work is original and of great interest. However, the quality of the ms. is rather low in terms of ms. organization, clarity of experimental design and presentation of the results. The main flaw of the work is that fragmentation is apparently determined only on the basis by microscopical observation of particle micro and nano fluorescent particles. However, the presentation and discussion of the results relies on precise quantitative data.	We thank the Reviewer for their critical and constructive comments on this manuscript with regard to the limitations of our methodology. Please see response to comment 28 and 36 below.

	Therefore, due to the importance of the results, the possible limitations of this approach should be considered.	
28	1) How can be the Authors sure that all the particles measured were plastic items? I can understand that FTIR could not work on the samples, but counting and measuring nano and micro-sized green fluorescent dots in a marine invertebrate fed on algae is a difficult task.	See response to reviewer 1- comment 23. Further, this method of counting ingested microplastics based on fluorescence is common in many papers E.g. Cole et al 2014, Isolation of microplastics in biota-rich seawater samples and marine organisms, Cole and Galloway 2016, Ingestion of Nanoplastics and Microplastics by Pacific Oyster Larvae, and Weber et al 2017, PET microplastics do not negatively affect the survival, development, metabolism and feeding activity of the freshwater invertebrate Gammarus pulex. Pilot studies revealed that the undigested algae did not fluoresce at the same wavelength or intensity as the plastics and was easily distinguishable from the plastic fragments. Further, control krill were treated the same way as experimental krill i.e. fed algae and these were imaged in the same manner as the experimental krill
29	2) How many samples for each type of determination were utilized? In the figures or text there is no mention of the number of samples on which statistical analysis was performed. For image analysis (Fig.2) apparently only 2 krill were utilized. How many for the homogenates and fecal pellets? These data must be clearly indicated.	Fig. 1 Caption added '...exposed to a Low (20%) (n= 9) and High (80%) (n=9) plastic concentrations' Fig 2. Caption added 'N=17 krill were used for histological analysis' Fig.3 Caption added 'A) 24h on Day 1 (n=6) and Day 4 (n=6), B) 4 (n=6) and 24h (n=6) on Day 4 only. All faecal material per beaker (5 krill) was pooled to form a single sample per time point. Line 362: Added 'Faecal material was pooled per beaker per time point resulting in 15 samples.' Further see Methods section: Line 334: Four day feeding and egestion experiments were carried out on 45 Antarctic krill, and Line 344: Three krill from each beaker were randomly selected for body burden analysis (n=18 krill) Line 349: To investigate tissue

		localisation of ingested plastic, two krill from each beaker were randomly selected, fixed in formalin, and used for histological cryo-section (20µm) analysis Line 356: These 5 krill were all used for tissue localisation analysis Line 377: 163 images were analysed No statistical analysis was carried out on particles identified in histological sections. This was qualitative data.
30	3) In the methods section, what does 'Particle Size Bioassay' mean?	Line X: Changed to 'Particle Size Experiment'
31	This weakness is apparently reflected in Abstract and introduction (last para), where great emphasis is given on the results obtained without clearly explaining HOW they were obtained.	Added to Abstract Line 29: 'However, little is known about the degradation of microplastics through ingestion. Here, by exposing Antarctic krill (Euphausia superba) to microplastics under acute static renewal conditions, we show the first observations of physical size alteration to microplastics ingested by a planktonic crustacean.' Further see response to Author 1 comment 12 and 14.
	Results	
32	This section is not well organized and therefore not clear. Results and discussion are mixed up in almost all the subsections. The sub-heading should be revised trying to group the main results obtained in whole homogenates and fecal pellets.	Not adopted: We thank the reviewer for their suggestion but the sections have been organised by type of analysis carried out and main findings they represent. Multiple samples types were used to draw the same conclusion. In our opinion it would be less clear to separate out the results by sample type, particularly as the samples types were often analysed in several ways to draw different conclusions. For example whole krill homogenates were analysed with both doses pooled to look as the overall trend of fragmentation vs whole beads. Whole krill homogenates were also compared between doses to look at incomplete fragmentation as a result of exposure. These analyses answer two different questions and we feel they should not be grouped together. See response to Reviewer 1 comment 13,16,17 and 18 for sections that have been rearranged to increase clarity of the manuscript
33	please refer to plastic 'suspension' and not 'solution' throughout the ms...	The word solution has been changed to suspension
34	What is more, in this section it should be	Further details have been added to the

	more clearly explained on how many samples measurements were performed and how the size was determined, if not within the text at least in figure legends. Instead of discussing the interpretation of the results, the way data were obtained should be clearly indicated.	Methods section and the Figure legends. Please see a full description in comment 29.
	Discussion	
35	The discussion should contain a critical and comparative evaluation of the results reported, before discussing their implications and significance .	We thank the reviewer for their suggestion and have revised the discussion to include discussion of the results.
36	A critical discussion of the methods used and on the reliability of the results obtained is needed.	Please see Line 196-200: for polymer limitations Line 232-247: for a critical discussion of methods. Supplementary Material section: Microplastic Characterization for Image analysis limitations. Data derived from the histological sections was qualitative. We were unable to get reliable size results for the nanoparticles in the digestive gland however, a particle must be smaller than 0.144um to pass through the primary filter into the digestive gland
37	Quantitative data must be compared with other studies reporting data on microplastic ingestion in terms of body burden and elimination with fecal pellets in other planktonic species.	The current study focuses on the way ingestion by Antarctic krill alters microplastics. Quantitative data regarding body burden and rates of elimination for microplastics in faecal pellets was outside the scope of this study. However the authors do have this type of data available in a similar study under review for publication elsewhere. This study was specifically designed to answer questions on ingestion, body burden and egestion and provide quantitative data for comparison with other planktonic crustaceans. This study will be publicly available shortly. The addition of quantitative data on ingestion, body burden and egestion rates would not change the major findings or key points raised in the current study: for the first time Antarctic krill (and potentially other species) are noted to be part of the degradation process of plastic in the ocean, and exposure concentration will affect the efficacy of fragmentation.
38	The choice of the Antarctic species should be justified.	Antarctic krill were selected as study species because they are the world's most abundant metazoan species and

		have highly flexible diet, yet we lack of microplastic studies with an Antarctic focus. Which we believe is a crucial knowledge gap in the current literature. Please see line 66: Added Line 68:
39	Moreover, since the concentrations of microplastic utilized were extremely high, and, fortunately, not in a predictable environmental range, the results should be also discussed in terms of the possible occurrence of fragmentation of microplastic by krill in real environmental conditions at much lower particle concentrations.	In light of the limited data presented in this study and in others reporting plastics in the southern ocean, the authors would prefer not to discuss this topic too heavily – to avoid making erroneous conclusions before sufficient data is available. A short discussion is presented in line 212-220: “ we suggest that current concentrations may be within the bounds of optimal trituration for krill, but fragmentation efficiency may be affected by chronic exposure. The increased fragmentation of plastic noted at low exposure conditions gives further weight to our hypothesis that digestive fragmentation is more common in the environment than recorded in current literature, which often use similarly high exposure concentrations for exposure experiments. Current contamination levels in the Southern Ocean are theoretically low enough to promote efficient digestive fragmentation by krill species, and in a global context, possibly for other zooplankton with sufficiently developed gastric mills”
	Reviewer #3 (Remarks to the Author): This is an interesting study and well presented, I believe it to be worthy of publication. The major claim of the paper is that ingestion of microplastic by krill results in fragmentation of the plastic to smaller nanoplastics. The study suggests that exposure to a high concentration of beads or repeated exposure decreases fragmentation. These finding are novel and I believe will be of interest to many researchers in the field of microplastic research and zooplankton ecology. What is emerging from the growing literature describing microplastic abundance is the mismatch between how much plastic is entering the oceans, and how much is found floating on the ocean surface. Models have conservatively estimated that	We thank the reviewer for their positive feedback on our manuscript. The reviewer makes a valid point regarding the weakness of this study. As the reviewer acknowledges, it was beyond the scope of this study to test other polymers. However testing on environmentally degraded mixture of polymers is the next logical step for this research. See lines 196-200 for a short discussion on this limitation

	over 5 trillion pieces of plastic are floating in the world's oceans and yet this accounts for approximately 1% of the estimated ocean plastic load. This raises the key question; where is the remaining 99% of plastic going; which this study begins to answer? I believe the conclusions of the paper are original and will influence thinking in the field of microplastic research. This study will contribute to our understanding the fate of microplastics in the marine environment and their potential impact on marine biota. A weakness of the paper is that the experiments have only been run with one polymer type (Polyethylene), the conclusions of the paper would be considerably strengthened if different polymer plastics as well as weathered and virgin plastics were used, however, I accept this is outside the scope of this study. In summary I think the manuscript is worthy of publication following minor revision, see marked manuscript. The study is reproducible from the methods given and will make a good contribution to this topical and important growing area of research.	
40	Line 47 add 'be'	Line 49 Added: Planktonic suspension and filter feeders may be the most susceptible to microplastic ingestion due to the relatively indiscriminate nature of this feeding strategy
41	Line 65 add 'to'	Line 90 Added: we exposed krill to a 4 day static renewal assay, which incorporated daily feeding on two (low - 20% and high - 80%) PE microplastic and algal diets.
42	Line 78 delete 'in'	Line 98 Changed: whereas the mean particle size isolated from within the krill was, on average, 78% smaller than the original beads
43	Line 93 change 'polymers' to 'polymer' and delete 'are'	Line 186 changed: Regardless of their original polymer properties, marine microplastics are largely comprised of secondary plastics, derived from the breakdown of larger plastic items
44	Line 108 insert 'to'	Line 111 added: Particles from the krill and bead blanks were found to be unaffected by the enzyme digestion protocol
45	Line 115 remove commas after 'both' and 'homogenates'	Line 117 This sentence has been restructured: To further explore this observation we compared the proportion of fragments to whole beads isolated from whole krill homogenates and faecal pellets exposed to the high and low treatment

46	Reference 7, Cole et al., is a lab based study. Used here it suggests microplastics isolated from wild caught zooplankton.	Reference was removed
47	Line 219 insert 'doing'	Line 265 added: This study uncovered the ability of an Antarctic keystone species to physically change ingested microplastics in a manner not previously described and in doing so, provides evidence for biologically facilitated production of nanoplastics
48	Line 289 delete 'of'	Line 347 changed: As the beads were fragmented after ingestion, the total bead ingestion rates could not be calculated from stomach content or egested material
49	Line 304 change 'switch' to 'switched'	Line 362 changed: After 10 days, the diet was switched to 100% algae for five days

REVIEWERS' COMMENTS:

Reviewer #1 (Remarks to the Author):

I think the main points raised in my previous round of review have been satisfactorily addressed. As such, I recommend the publication of this manuscript in Nature Communications.

Reviewer #3 (Remarks to the Author):

I think the paper has been greatly improved following the revisions suggested by the reviewers. Following minor corrections (see attached), and a thorough proof read for small errors I believe the manuscript should be accepted for publication.

Editorial Note: The manuscript file marked up by Reviewer #3 a part of this review is unable to be reproduced as part of the peer review file.

	Reviewer Comments Reviewer 3	Response
1	I think the paper has been greatly improved following the revisions suggested by the reviewers. Following minor corrections (see attached), and a thorough proof read for small errors I believe the manuscript should be accepted for publication	
2	Line 30 insert comma and lowercase h	Line 30'... research, however,...'
3	Line 55 add s	Line 55 'Detrimental health effects...'
4	Line 57 delete possible	Line 57 '....carries a threat of plastic bioaccumulation....'
5	Line 72 add are	Line 72 '...and are then transported to the mandibles for mastication.'
6	Line 79 delete extra bracket	
7	Line 92 (low - 20% and high - 80%) This needs clarifying.	Line 93 added 'A 'low' diet was comprised of 20% plastic and 80% algae, the 'high' diet was comprised of 80% plastic and 20% algae.'
8	Line 155 delete 'did not' add d	Line 155 '...digestive enzymes contributed....'
9	Line 182 delete of which	Line 182 '...many possess similarly developed gastric mills...'
10	Line 316 delete ;	Line 316 '....(n=65, wet weight: 0.556 ± 0.117mg, length: 41.1 ± 3.7mm) were acclimatised....'
11	Line 370 capitalise milli-q	Line 370 '...Milli-Q..'
12	Line 420 delete extra supported by	Line 420 'A.D was supported by PHD scholarship...'